# Non-invasive optical control of endogenous Ca$^{2+}$ channels in awake mice

Sungsoo Kim [1,7], Taeyoon Kyung[2,6,7], Jae-Hee Chung[1], Nury Kim[2], Sehoon Keum[2], Jinsu Lee[1], Hyerim Park[1], Ho Min Kim[3,4], Sangkyu Lee[2]*, Hee-Sup Shin[2]* & Won Do Heo [1,2,5]*

Optogenetic approaches for controlling Ca$^{2+}$ channels provide powerful means for modulating diverse Ca$^{2+}$-specific biological events in space and time. However, blue light-responsive photoreceptors are, in principle, considered inadequate for deep tissue stimulation unless accompanied by optic fiber insertion. Here, we present an ultra-light-sensitive optogenetic Ca$^{2+}$ modulator, named monSTIM1 encompassing engineered cryptochrome2 for manipulating Ca$^{2+}$ signaling in the brain of awake mice through non-invasive light delivery. Activation of monSTIM1 in either excitatory neurons or astrocytes of mice brain is able to induce Ca$^{2+}$-dependent gene expression without any mechanical damage in the brain. Furthermore, we demonstrate that non-invasive Ca$^{2+}$ modulation in neurons can be sufficiently and effectively translated into changes in behavioral phenotypes of awake mice.

[1] Department of Biological Sciences, Korea Advanced Institute of Science and Technology (KAIST), Daejeon, Republic of Korea. [2] Center for Cognition and Sociality, Institute for Basic Science (IBS), Daejeon, Republic of Korea. [3] Center for Biomolecular and Cellular Structure, Institute for Basic Science (IBS), Daejeon, Republic of Korea. [4] Graduate School of Medical Science & Engineering, Korea Advanced Institute of Science and Technology (KAIST), Daejeon, Republic of Korea. [5] KAIST Institute for the BioCentury, Korea Advanced Institute of Science and Technology (KAIST), Daejeon, Republic of Korea. [6] Present address: David H. Koch Institute for Integrative Cancer Research, Massachusetts Institute of Technology, Cambridge, MA, USA. [7] These authors contributed equally: Sungsoo Kim, Taeyoon Kyung. *email: sklee@ibs.re.kr; shin@ibs.re.kr; wondo@kaist.ac.kr

The brain utilizes a versatile $Ca^{2+}$-signaling toolkit to regulate manifold functions including memory, emotion, and locomotion[1,2]. Accumulating evidence suggests that abnormally modulated intracellular $Ca^{2+}$ dynamics are correlated with brain dysfunctions such as neurodegenerative diseases[3–5], but their causality and specific contribution of $Ca^{2+}$ signaling per se on functional outcomes still remains elusive. Thus, to understand exact roles of neural $Ca^{2+}$ signaling in brain functions, it is necessary to specifically control intracellular $Ca^{2+}$ dynamics at designed time and space. The ubiquitously expressed $Ca^{2+}$-release-activated $Ca^{2+}$ (CRAC) channel is known to selectively introduce $Ca^{2+}$ inside the cells and has been targeted for the development of optogenetic $Ca^{2+}$ modulators[6–10], which were typically constructed from cytosolic domains of STIM1 (Stromal interaction molecule 1) protein, a CRAC channel regulator[11], and blue light-responsive plant photoreceptors, such as cryptochrome2 (CRY2)[12] or the LOV2 (light-oxygen-voltage-sensing) domain[13]. These optogenetic tools are able to shape distinguished intracellular $Ca^{2+}$ dynamics in response to various inputs of light and effectively evoke $Ca^{2+}$-responsive diverse biological events, including gene expression, cell migration, immune responses, and memory reinforcement. However, it has been found that excessive expression of designed elements alters basal $Ca^{2+}$ concentration, even in the dark state[6,7]. In addition, it is presumed that these optogenetic modules inevitably necessitate the use of optic fibers for in vivo brain applications because of the poor tissue-penetration efficiency of blue light[14,15]. Long-term insertion of optic fiber in mice brain also introduces biocompatibility issues, which induces astrocytic scar formation nearby implanted region[16,17], thermal damage[18,19] as well as morphological changes in neurons[20]. Recent studies have sought to address this challenge using upconversion nanoparticles (UCNPs), which emit visible light upon absorption of tissue-penetrating near-infrared (IR) light, to activate blue light optogenetic modules in vivo with non-invasive light stimulation[8,21]. However, this approach is not readily accessible to most users because of its requirement for the synthesis of UCNPs and the need for additional optimization of particle dosing and parameters of light illumination on target sites.

Here, we describe an optogenetic $Ca^{2+}$ modulator with ultralight sensitivity that can be readily activated in vivo awake mouse brain through non-invasive light illumination. We demonstrate fundamental properties of our advanced $Ca^{2+}$ modulator with rationally designed CRY2 mutant and find more improved characteristics, including minimal alteration of basal $Ca^{2+}$ level and higher responsiveness to light. Notably, we achieve induction of $Ca^{2+}$-responsive gene expression in both neurons and astrocytes in deep-brain regions including hippocampal or thalamic regions via custom-designed, non-invasive light-delivery system. Moreover, we show non-invasive neuronal $Ca^{2+}$ activation in different brain areas can lead to distinguished behaviors, suggesting broad applicability of our optogenetic method to modulate $Ca^{2+}$-responsive brain functions in vivo.

## Results

**Engineering *At*CRY2 to improve properties of OptoSTIM1.** Previously we demonstrated that oligomerization of OptoSTIM1[6], optogenetic $Ca^{2+}$ modulator, through blue light-driven CRY2 homo-association was able to efficiently activate endogenous CRAC channels, in turn elevating intracellular $Ca^{2+}$ concentration (Fig. 1a). But under excessive expression of the fusion protein, some cells exhibited slight increase of basal $Ca^{2+}$ concentration[6], possibly raising concerns of changed cellular contexts regardless of light illumination. We hypothesized that the dimeric nature of both CRY2[22] and STIM1[23,24] would

contribute to the multimeric property of OptoSTIM1 without blue light, promoting constitutive $Ca^{2+}$ influx through CRAC channels and leading to elevated basal $Ca^{2+}$ level. On the basis of previous findings that CRY2 is the main determinant of the kinetic properties of OptoSTIM1[6], we reasoned that disrupting the dimeric interface of CRY2 would reduce light-independent self-association of OptoSTIM1 and prevent subsequent CRAC channel leakage. To investigate the putative dimerization region of *Arabidopsis thaliana* CRY2 (*At*CRY2), we aligned the amino-acid sequence of *At*CRY2 with those of relatively well-characterized cryptochromes from other species and predicted its structure based on that of *At*CRY1 (Supplementary Figs. 1 and 2). Because the protrusion loop (Phe288–Ala306) in *Drosophila* cryptochrome[25] was reported to stabilize self-association through disulfide bond formation involving Cys296, we first located an equivalent protrusion loop (Lys268–Leu286) in *At*CRY2 model structure. Then, we selected the five amino acids, $N_{277}SEGE_{281}$, which potentially makes the close contact between two protrusion loops of *At*CRY2 (Fig. 1b). We then designed OptoSTIM1 variants in which the negatively charged glutamic acid residues Glu279 and Glu281 in the protrusion loop were replaced with the neutral amino-acid alanine (E279A and E281A) to reduce their potential electrostatic interaction, or in which the small residue Gly280 was converted into a bulky Tyr residue (G280W) to generate steric repulsion during self-association. OptoSTIM1 variants containing G280W or E281A mutations in *At*CRY2 exhibited lower basal intracellular $Ca^{2+}$ concentrations ($[Ca^{2+}]_i$) and similar maximal $[Ca^{2+}]_i$ in response to activation, measured by the ratiometric $Ca^{2+}$ indicator, Fura-2, compared with OptoSTIM1 bearing original CRY2 (CRY2) (Fig. 1c and Supplementary Fig. 3a, b). This suggests that Gly280 and Glu281 might be involved in CRY2 self-association in the dark, but not in the light-driven activation of OptoSTIM1. Notably, the correlation between basal $[Ca^{2+}]_i$ and protein expression level was lower for OptoSTIM1 carrying either $CRY2^{G280W}$ or $CRY2^{E281A}$ mutants compared with those bearing CRY2 or $CRY2^{E279A}$ (Supplementary Fig. 3c). Moreover, OptoSTIM1($CRY2^{E281A}$) produced a higher maximal $[Ca^{2+}]_i$ following stimulation compared with other variants, but had activation ($Ta_{1/2} = 62$ s) and deactivation ($Td_{1/2} = 7$ min) kinetics similar to those of the original OptoSTIM1. In contrast, OptoSTIM1($CRY2^{G280W}$) exhibited lower maximal $[Ca^{2+}]_i$, slower activation, and faster deactivation compared with the original OptoSTIM1 (Fig. 1d and Supplementary Fig. 4).

To further enhance the photosensitivity of OptoSTIM1 ($CRY2^{E281A}$), we employed the previously reported superior CRY2-clustering systems, CRY2olig (E490G)[26] and CRY2clust (A9)[27], where C-terminal amino-acid sequence of CRY2 is extended with nine residues (ARDPPDLDN). Combining $CRY2^{E281A}$ with either $CRY2^{E490G}$ or A9 minimally affected $[Ca^{2+}]_i$ in both dark and light conditions beyond that displayed by OptoSTIM1($CRY2^{E281A}$) (Fig. 1c and Supplementary Fig. 3a, b). Interestingly, compared with other variants, OptoSTIM1 ($CRY2^{E281A}$-A9) exhibited much higher sensitivity to light in that ~47% of OptoSTIM1($CRY2^{E281A}$-A9)-expressing cells remained effectively responsive to a light intensity of $1\,\mu W\,mm^{-2}$, and showed ~55-fold higher sensitivity than original OptoSTIM1 (Fig. 1e–g and Supplementary Figs. 6, 7). In contrast, OptoSTIM1 ($CRY2^{E281A,E490G}$) showed lower responsiveness to light and slower deactivation kinetics. OptoSTIM1 variants showed differential activation and deactivation kinetics in terms of $[Ca^{2+}]_i$ dynamics, probably owing to each mutation rendering distinguished clustering characteristic of CRY2. Interestingly, they consistently exhibited faster activation and slower deactivation kinetics upon elevated light density (Supplementary Fig. 8). In these characterizations, we utilized R-GECO1 as an

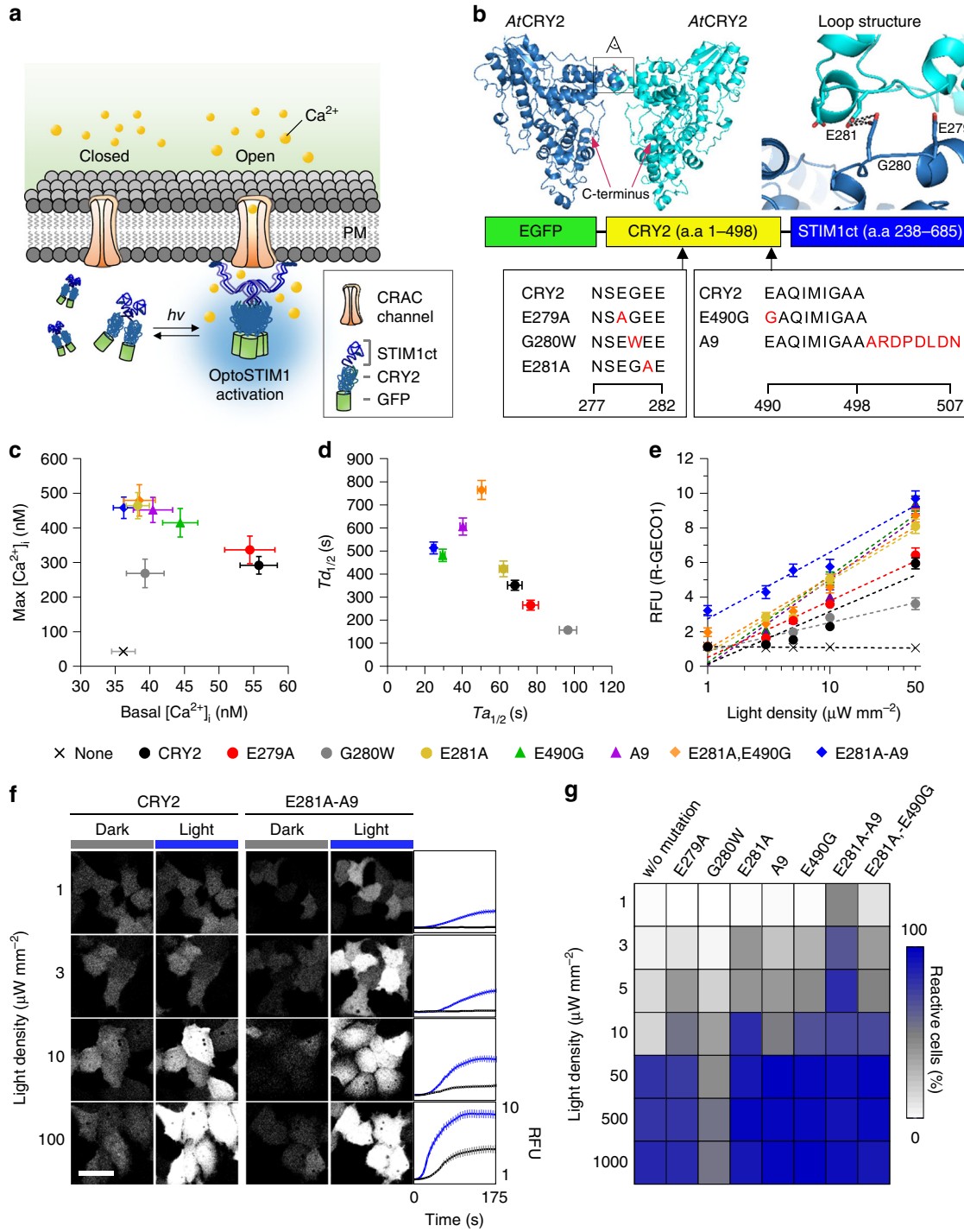

**Fig. 1 Development of ultra-light-sensitive OptoSTIM1. a** Schematic representation of working mechanism of the OptoSTIM1. PM, plasma membrane; STIM1ct, C-terminal fragment of STIM1 (a.a 238−685). **b** Model structure of *At*CRY2 predicted by SWISS-MODEL. Expanded view on the right represents loop structure in the potential dimeric interface highlighted in black box on the crystal structure; amino acids for the target region for mutagenesis are aligned. Reds indicate specific residues mutagenized. **c** Graph showing correlation between $[Ca^{2+}]_i$ in dark (*x* axis) and $[Ca^{2+}]_i$ upon light stimulation (*y* axis) measured using Fura-2. **d** Plot representing correlation between half-maximal time point for reaching saturated R-GECO1 level upon light stimulation (*x* axis) and basal R-GECO1 level in dark (*y* axis) for each indicated variant ($n \geq 100$ for each variant). **e** Graph showing maximal R-GECO1 fluorescence intensity based on varying light density (1−50 μW mm$^{-2}$). Blue light was delivered at each indicated power density at 5-second intervals for 1 minute using a 488 nm laser ($n \geq 100$ for each variant at each light density). **f** Representative R-GECO1 images of cells expressing either OptoSTIM1(CRY2) or OptoSTIM1(CRY2$^{E281A}$-A9) (488 nm, 1−100 μW mm$^{-2}$, 5-second intervals for 1 minute). Graphs on the right indicate fluorescence change of R-GECO1 upon light stimulation (Black, CRY2; Blue, CRY2$^{E281A}$-A9). Scale bar, 10 μm. **g** Heat plot showing extent of reactive cell population upon light stimulation with each indicated light density. Images and quantified data are representative of multiple experiments ($n > 3$). Data represent means ± s.e.m.

intensiometric $Ca^{2+}$ biosensor. As the previous study showed that R-GECO1 itself can be activated by blue light and elicit change of fluorescence[28], we examined if there is any unintended additive effect by R-GECO1 on our quantified results. Notably, HeLa cells solely expressing R-GECO1, showed subtle change (<3%) of fluorescence intensity upon continuous or transient exposure to blue light, which is much smaller than typical dynamic range (>600%) of fluorescence change we observe through our optogenetic $Ca^{2+}$ modulators (Supplementary Fig. 9). Therefore, this result indicates that photoactivation of R-GECO1 had a negligible effect on our experimental condition. From all the results above, we conclude that OptoSTIM1(CRY2$^{E281A}$-A9) showed performance and characteristics closest to the goal of our CRY2 engineering effort, exhibiting lower basal $[Ca^{2+}]_i$ regardless of expression level with superior light responsiveness for effective induction of $Ca^{2+}$ influx. We termed this ultra-photosensitive variant OptoSTIM1(CRY2$^{E281A}$-A9) as monster-OptoSTIM1, abbreviated as monSTIM1.

**Characteristics of monSTIM1**. As previously described, fundamental properties of OptoSTIM1 include plasma membrane translocation upon light stimulation and possessing a specific window of photoactivatable light spectrum[6] (400–500 nm) for compatible use with red-shifted biosensors. To verify whether monSTIM1 retains these properties, first, we closely examined the change of subcellular distribution of monSTIM1 during the process of light irradiation. By utilizing plasma membrane marker protein, iRFP670-PM(KRas4B tail), we clearly show translocation of light-activated monSTIM1 to the plasma membrane (Supplementary Fig. 10a–c). Second, to investigate light spectrum for monSTIM1 activation, we stimulated monSTIM1-expressing cells with different wavelengths of light. We observed that monSTIM1 efficiently responds to 457 or 488 nm light, but weakly or does not respond to the 405 nm and wavelength of light longer than 514 nm (Supplementary Fig. 10d) equivalent to what OptoSTIM1 has previously shown. Among distinguished features between OptoSTIM1 and monSTIM1, next we particularly focused on to address what drives elevated basal $[Ca^{2+}]_i$ by OptoSTIM1 compared with that of monSTIM1, which might change the physiological context of cells under overexpressed condition. We hypothesized that dimeric feature of both SOAR (a.a 336–485) domain in the C-terminus of STIM1 and CRY2 would contribute to induction of higher oligomeric state of OptoSTIM1 in the absence of blue light and, in turn, basal $[Ca^{2+}]_i$ increment, and CRY2$^{E281A}$ mutation on monSTIM1 would reduce the degree of oligomeric state in the dark. To address whether CRY2$^{E281A}$ mutation indeed results in the change of basal oligomeric state, we compared homo-association property of OptoSTIM1 and monSTIM1 by utilizing InCell SMART-i (Intracellular supramolecular assembly readout trap for interactions)[29], which readily assesses protein interactions in the form of cluster formation (Supplementary Fig. 11a). We monitored cluster formation upon rapamycin treatment in HeLa cells co-expressing FKBP-$V_H$H (GFP), FRB-mScarlet-FT (Ferritin) with EGFP-STIM1ct (a.a 238–685), OptoSTIM1 (EGFP-CRY2-STIM1ct), or monSTIM1 (EGFP-CRY2$^{E281A}$-A9-STIM1ct) both in the dark and light. Relatively smaller cell population with cluster formation was visualized with cells expressing either EGFP-STIM1ct (14.2%) or monSTIM1 (11%), whereas 39.2% of OptoSTIM1-expressing cell population showed cluster formation (Supplementary Fig. 11b, c). Light-stimulated cells expressing either OptoSTIM1 or monSTIM1 showed robust and comparable level of cluster formation inside the cells, owing to homo-oligomerization of CRY2. This result suggests that CRY2$^{E281A}$ would have less-oligomeric property than CRY2 in the absence of blue light, thereby

significantly attenuating elevated basal $[Ca^{2+}]_i$ even under over-expressed condition.

**Utilizing CRY2$^{E281A}$-A9 to regulate receptor tyrosine kinase**. In order to examine applicability of the light-sensitive CRY2$^{E281A}$-A9 module to other CRY2-based optogenetic tools, we employed OptoFGFR1 (optical activation of fibroblast growth factor receptor 1)[30] and replaced original CRY2 with CRY2$^{E281A}$-A9 (Supplementary Fig. 12a). In this case, we barely detected differences in both basal $[Ca^{2+}]_i$ and light-induced maximal $[Ca^{2+}]_i$ between cells expressing either OptoFGFR1 or OptoFGFR1(CRY2$^{E281A}$-A9). We suspect that this phenomenon is due to inherent working mechanism of FGFR1 in elevating $[Ca^{2+}]_i$, which triggers transient IP$_3$-induced $Ca^{2+}$ ion release from intracellular stores, such as ER (Supplementary Fig. 12b), where we might have saturated activity of OptoFGFR1 in this particular light-illuminating condition. However, upon titrating exposure time of light, we found that 76% of cells expressing OptoFGFR1(CRY2$^{E281A}$-A9) was reactive to 1.5 s exposure of light, whereas only 15% of cells expressing OptoFGFR1 responded (Supplementary Fig. 12c, d), further demonstrating the versatility of CRY2$^{E281A}$-A9 to generate light-sensitive optogenetic module.

**Non-invasive monSTIM1 activation in vivo**. This ultra-photosensitivity led us to anticipate that monSTIM1 might be able to drive $Ca^{2+}$-dependent molecular activity in neurons of the intact brain in response to non-invasive light illumination in a similar way that we reported previously[31]. To test this conjecture, we designed a light-illuminating cage in which an LED solid-state array capable of delivering blue light (473 nm) of ~1 mW cm$^{-2}$ to the head of mice was attached to the cage lid (Fig. 2b and Supplementary Fig. 13). In this condition, we could not find any noticeable change in mice behavior by light illumination per se (Supplementary Movie 1). Then, excitatory neurons of the somatosensory cortex (S1) were transduced with lentiviral constructs of OptoSTIM1, monSTIM1, or light-insensitive OptoSTIM1(CRY2$^{D387A}$). After 4 weeks, mice were exposed to blue light for 30 minutes at homecage without removal of hairs and skin, and then killed 60 minutes later to assess for the expression level of c-Fos, a $Ca^{2+}$-dependent immediate-early gene (Fig. 2a). Strikingly, mice expressing monSTIM1 showed a significant induction of c-Fos expression compared with control groups including mice in ambient room light condition without blue light illumination and mice expressing light-insensitive OptoSTIM1(CRY2$^{D387A}$) with blue light exposure for 30 minutes. We found the c-Fos signal was predominantly localized to cells expressing monSTIM1 (Fig. 2c, d). We next sought to validate the functionality of monSTIM1 in astrocytes, another major cell type in the brain. Activation of monSTIM1, expressed under the control of the *GfaABC1D* promoter, efficiently induced expression of c-Fos in astrocytes. Specifically, c-Fos was detected in 74% of the monSTIM1-positive astrocyte population in the S1 region, whereas control groups showed no noticeable c-Fos induction. To evaluate the suitability of monSTIM1 for deep-brain modulation, we examined c-Fos induction in monSTIM1-expressing astrocytes in the dentate gyrus (DG) and thalamic (TH) regions of the brain under the same light-stimulation condition we used for cortical stimulation. We found that 57% (DG) and 44% (TH) of cell population expressing monSTIM1 showed c-Fos expression upon light stimulation (Fig. 2e, f). The decreased percentages of c-Fos-positive astrocytes in DG and TH compared with that of S1 reflect that penetration efficiency of blue light was gradually reduced as a function of depth in the brain. We also observed that 21.5% of monSTIM1-positive excitatory neurons in the

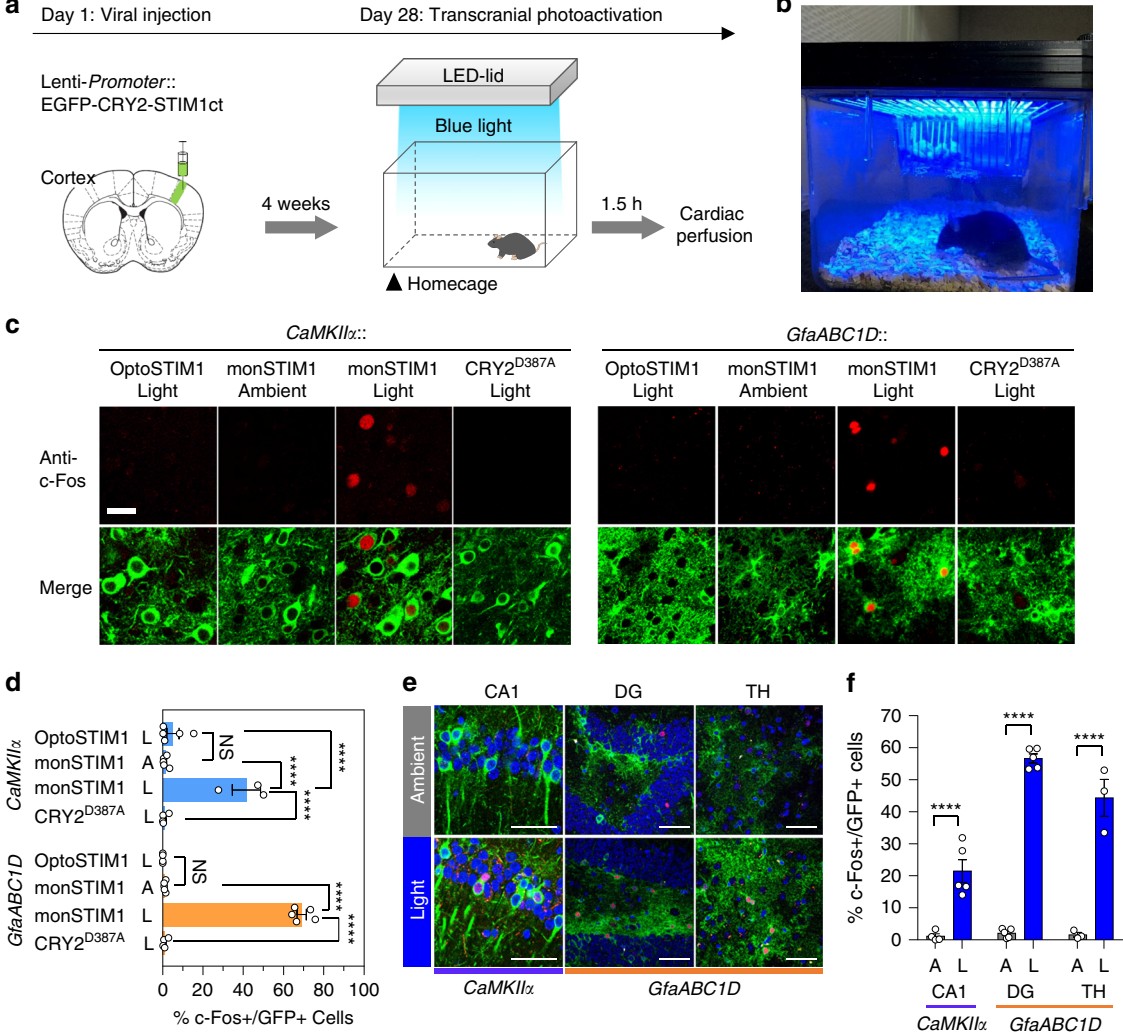

**Fig. 2 Optogenetic Ca$^{2+}$ modulation in the brain through non-invasive light delivery. a** Schematic representation of the procedure for variable OptoSTIM1 expression and activation by non-invasive light stimulation. Lentivirus packaged with different OptoSTIM1 variants expressed under the control of either a *CaMKIIα* promoter (excitatory neurons) or *GfaABC1D* promoter (astrocytes) targeted to the S1 cortical region. Four weeks post injection, mice were illuminated with LED light in their homecage and subsequently killed. **b** Customized transcranial light illumination system. A solid-state LED is attached to the cage lid, and its light intensity is controlled by a panel. **c** Representative images showing c-Fos–positive cells expressing each OptoSTIM1 variants. Scale bar, 20 μm. **d** Summary plot showing quantified population of c-Fos–positive (+) cells expressing OptoSTIM1 variants (****$P < 0.0001$; Sidak's tests). **e** Representative images of c-Fos-stained cells with or without monSTIM1 activation in designated brain regions. Blue, DAPI; Red, c-Fos; Green, monSTIM1. Scale bar, 50 μm. **f** Graph showing the percentage of c-Fos–positive cells expressing monSTIM1 in **e**. (****$P < 0.0001$; Sidak's tests). Images and quantified data are representative of multiple experiments ($n > 3$). Data represent means ± s.e.m.

hippocampus CA1 region showed c-Fos expression. Therefore, these results demonstrate that monSTIM1 is able to effectively induce intracellular Ca$^{2+}$ signaling in deep-brain regions through non-invasive light activation.

**Modulating mice behaviors through monSTIM1.** At last, we explored whether induction of Ca$^{2+}$ signaling through non-invasive light delivery impacted specific behaviors of awake mice. We have previously demonstrated that the activity of voltage-dependent L-type Ca$^{2+}$ channels (Ca$_v$1.2, *Cacna1c*) in the anterior cingulate cortex (ACC) is involved in social fear learning in mice (e.g., observational fear response)[32]. Mice with an ACC-limited deletion of the Ca$_v$1.2 gene showed reduced observational fear, likely owing to impaired synaptic transmission or neuronal excitability. In addition, previous studies demonstrated that Ca$^{2+}$ signaling is involved in electrophysiological properties of

excitatory neurons of the ACC. Ca$^{2+}$-stimulated proteins such as Ca$_v$1.2, CaM, CaMKIV and AC1[33] are known to contribute to the expression of immediate-early genes, thereby promoting long-term potentiation in ACC[34]. However, the causal relationship between direct activation of Ca$^{2+}$ signaling and observational fear has not been tested. Accordingly, we targeted excitatory pyramidal neurons in the ACC with EGFP, monSTIM1 or light-insensitive OptoSTIM1 and examined socially transmitted fear responses (Fig. 3a). In these experiments, only the observer mouse was illuminated for 30 minutes in the homecage and then both observer and demonstrator mice were moved into observational fear behavioral chambers separated by a transparent Plexiglas partition. Mice were habituated to the apparatus for 5 minutes, and repetitive foot shocks were applied only to the demonstrator at 10-second intervals for 4 minutes to evoke vicarious freezing response in the observer (Fig. 3b). Notably,

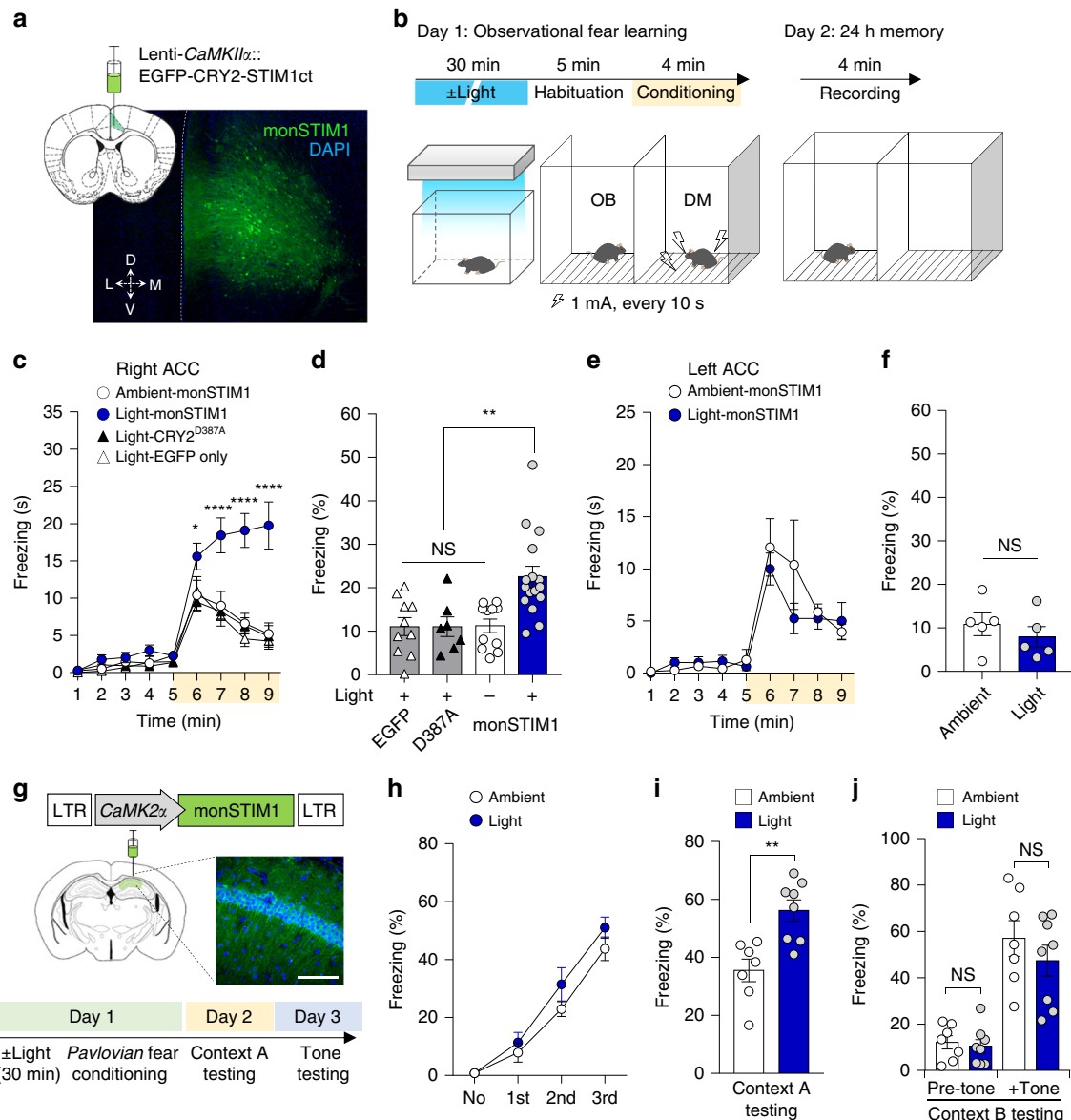

**Fig. 3 Effects of non-invasive optical Ca²⁺ modulation on specific mice behaviors. a** Image showing representative histology sample of right ACC expressing monSTIM1. **b** Schematic depiction of observational fear learning paradigm. OB, observer mouse. DM, demonstrator mouse. **c** Summary data showing average time for freezing. Yellow box indicates the time window for foot-shock administration. (*$P < 0.05$, ****$P < 0.0001$; $P$ value was determined by one-way ANOVA). **d** Graph describing the percentage of time spent freezing during the 24-hour memory test (**$P < 0.01$, NS = Not significant; Tukey's tests). **e** Graph showing average time for freezing with mice expressing monSTIM1 at left ACC on acquisition of observational fear (Day 1). **f** Graph indicating the percentage of time spent freezing during 24-hour memory test (NS = Not significant; Student two-tailed $t$ test). **g** Image showing fluorescence image of histology of right CA1 hippocampus (Blue, DAPI; Green, OptoSTIM1; Scale bar, 50 μm) with schematic depiction of conducted fear conditioning experiment. **h–j** Graphs showing the percentage of freezing behavior of mice at each training points during fear conditioning **h**, at 24-hour contextual memory test **i**, and at 48-hour post training with tone memory test **j**. (**$P = 0.002$ for light-illuminated versus ambient; NS = Not significant by Student's two-tailed $t$ test. Images and quantified data are representative of multiple experiments ($n > 3$). Data represent means ± s.e.m.

mice in which monSTIM1 was activated displayed significantly higher freezing levels during the training period (4 minutes). Twenty-four hours later, observers re-exposed to the same conditioning chamber also showed increased contextual fear memory compared with mice in the control groups (Fig. 3c, d), indicating that Ca²⁺ signaling in the ACC contributed to both short- and long-term social fear responses. In accordance with our previous study demonstrating functional lateralization in modulation of observational fear, we also observed lateralized brain functions of Ca²⁺ signals in observational fear[35], demonstrating that activation of Ca²⁺ signals by monSTIM1 only in the right, and not in the left, hemisphere resulted in observational fear response

(Fig. 3e, f). Given that activation of monSTIM1 in the ACC did not cause any alterations in locomotor activity or anxiety-like behavior during open-field tests (Supplementary Fig. 14), these results collectively demonstrate that monSTIM1-mediated Ca²⁺ induction in the ACC selectively enhances socially transmitted fear response in mice.

Next we explored whether non-invasive activation of monSTIM1 in the CA1 hippocampus, deeper brain area than the ACC, could still exhibit the functionality to modulate brain functions such as reinforcement of contextual memory as we previously described[6]. Mice expressing monSTIM1 were illuminated with blue light at the homecage and moved to conditioning

chamber to conduct Pavlovian fear conditioning administrating an auditory cue (90 dB, 3 kHz for 30 s) with pairing of an electric foot shock (2 s, 0.7 mA) (Fig. 3g). To test memory formation, mice were re-exposed to either the same context (day 2) or the tone (day 3) sequentially. Consistent with our previous finding, monSTIM1-stimulated mice exhibited enhanced context-specific memory compared with non-stimulated mice but no differences in tone memory, demonstrating that monSTIM1 activation through non-invasive light delivery sufficiently modulate specific brain function and corresponding behavioral outcome (Fig. 3h–j).

## Discussion

In this study, we present an ultra-light-sensitive optogenetic $Ca^{2+}$ modulator achieved with CRY2 engineering, which includes mutated single amino acid on predicted CRY2 dimeric interface ($CRY2^{E281A}$) and superior clustering module (CRY2clust (A9)). We demonstrate that high propensity of OptoSTIM1 to achieve advanced oligomeric state in the basal condition through homo-interaction of both CRY2 and STIM1ct drives elevation of resting $Ca^{2+}$ level. We also show that $CRY2^{E281A}$ mutant, when incorporated into OptoSTIM1 in place of CRY2, attenuates increased resting $Ca^{2+}$ level upon reduced propensity of oligomerization. Notably, monSTIM1 shows minimal alternation of basal $[Ca^{2+}]_i$ compared with OptoSTIM1 encompassing original CRY2 while retaining fundamental properties of OptoSTIM1, such as light-dependent plasma membrane translocation and a defined window of stimulation light spectrum (400−500 nm). Also, we demonstrate that $CRY2^{E281A}$-A9 has broad applicability in potentiating light sensitivity of other CRY2-based optogenetic tools. This advanced CRY2 module with enhanced light sensitivity can be more suitably utilized in vivo mouse brain upon non-invasive light delivery without necessity of chronic implantation of optic fiber, thereby greatly minimizing unintended mechanical effects on the brain. We demonstrate our custom-designed light-delivery chamber can sufficiently stimulate mon-STIM1 to induce subsequent $Ca^{2+}$ signaling in neurons and astrocytes of various brain regions even without shaving hairs of mice, whereas ambient room light condition, which we designated as ambient in this study could not show any effect. These non-invasive neuronal $Ca^{2+}$ modulation enable modulation of specific brain functions and corresponding animal behaviors. Activating monSTIM1 in excitatory neurons at right but not left ACC enhances socially transmitted fear, depicting importance of $Ca^{2+}$-mediated signaling cascade on arousing empathy-like behavior. In addition, non-invasive $Ca^{2+}$ increment at CA1 hippocampus influence contextual memory reinforcement, representing broad utilities of our optogenetic tool on deep-brain $Ca^{2+}$ studies. Even though we previously provided a piece of evidence that long-term activation (25 min) of our optogenetic calcium modulator had no significant effect on resting membrane potential[6], specific physiological properties changed by monSTIM1-mediated $Ca^{2+}$ signaling and critical time window for light illumination to eventually result in behavioral outcomes still remain elusive and are needed to be further characterized in the future study. We also speculate that there might be a threshold of duration of light illumination that increases intracellular $Ca^{2+}$ level for at least 25 minutes rather than timescale of seconds or a few minutes, which is required to eventually results in a significant change of behavioral outcomes. The identify of threshold might be the number of responded cells[36] or activation of a set of certain $Ca^{2+}$-responsive molecular machineries including activity-induced synaptic remodeling molecules, such as GluR and CaMKII, which are previously shown to influence behavioral effects upon their modulated activities[37–39]. In addition, it has been recently shown through chemogenetic tools that

AMPA and NMDA receptor activities can be gradually modulated over the course of 30 min post ligand stimulus in hippocampal excitatory neurons[40], indicating that $Ca^{2+}$-responsive molecular machineries may take timescale of minutes to impose their influences on higher level phenotypes.

Combined with approaches for expressing genes in the brain through systemic delivery of engineered viruses[41] and various transgenic lines for gene recombination, the approach demonstrated here should allow bona fide non-invasive expression and activation of $Ca^{2+}$ signaling in a broad range of cell types, allowing fundamental roles of $Ca^{2+}$ in manifold brain functions to be elucidated.

## Methods

**Sequence alignment and model building**. Sequences of cryptochrome proteins from various species were aligned using the web-based software MultAlin[42]. The model structure of $At$CRY2 was constructed by homology modeling using SWISS-MODEL software. $At$CRY1 structure (PDB code: 1U3D) was selected as templates for model building based on the calculated Global Model Quality Estimation (GMQE) score (>0.5), and the model with highest Qualitative Model Energy ANalysis-Z (QMEAN-Z, −1.63) was selected as the final $At$CRY2 model. This final $At$CRY2 model was structurally aligned to *Drosophila* cryptochrome dimer (PDB code: 4K03) to predict the potential dimeric interface of $At$Cry2.

**Plasmid construction**. Construction of expression plasmids for R-GECO1 and OptoSTIM1 were previously described in detail[6]. The OptoSTIM1 used in the current study encompassed the photolyase homology region of CRY2 (amino acids 1–498) and C-terminus of STIM1 (amino acids 238–685), both of which were subjected to mutagenesis through polymerase chain reaction (PCR)-driven overlap extension using forward and reverse mutagenic oligonucleotides for each and two flanking primers. Flanking primers and mutagenic oligonucleotides used for mutagenesis of CRY2 were as follows: Flanking primers, 5′-GTAACCGGTCAT-GAAGATGGACAAAAAGACCATCGTCTG-3′ (forward) and 5′-CTCCGCCTCCCCCACTGAATTCGGCAGCACCGATCATAATC-3′ (reverse); E279A, 5′-AACAGC**GCC**GGCGAAGAAAGCGCCGATCTGTTCCTG-3′ (forward) and 5′-GCTTT**CTTC**GCCGGCGCTGTTTTTATCGCGAGCCCA-3′ (reverse); G280W, 5′-AACAGCGAA**TGG**GAAGAAAGCGCCGATCTGTTCC-3′ (forward) and 5′-GCTTTCTTC**CCA**TTCGCTGTTTTTATCGCGAGCCCA-3′ (reverse); E281A, 5′-GAAGGC**GCC**GAAAGCGCCGATCTGTTCCTG-3′ (forward) and 5′-GCTTT**CGGC**GCCTTCGCTGTTTTTATCGCGAG-3′ (reverse).

The resulting PCR-amplified sequences encoding CRY2 were cloned into the OptoSTIM1(CRY2) vector at *Age*I and *Eco*RI sites or *Bsp*EI and *Bam*HI sites. OptoSTIM1 containing oligomeric mutants (CRY2olig (E490G) and CRY2clust (A9)) were constructed as previously described[17]. For construction of OptoSTIM1 bearing both $CRY2^{E281A}$ and oligomeric mutants (CRY2olig (E490G) and CRY2clust (A9)), a sequence encoding the CRY2(E281A) mutant was inserted into either OptoSTIM1(CRY2clust (A9)) or OptoSTIM1($CRY2^{E490G}$) at *Age*I and *Sac*I sites. Construction of expression plasmid for FKBP-V$_H$H(GFP) was previously described in detail[6]. FRB-mScarlet-Ferritin (FT) was constructed by cloning exchange mScarlet (pmScarlet-C1, plasmid # 85042; RRID:Addgene_85042) into FRB-EGFP-FT using *Age*I and *Bsr*GI sites. piRFP670-N1 was a gift from Vladislav Verkhusha (Addgene plasmid # 45457; RRID:Addgene_45457). The sequence encoding iRFP670 was inserted into pEGFP-C1 (Clontech) after excising EGFP with *Age*I and *Bsr*GI restriction enzymes to generate piRFP670-C1 vector. Next, the tail sequence of KRas4B tail (20 amino acids; KMSKDGKKKKKKSKTKCVIM) was fused to the piRFP670-C1 using *Bsr*GI and *Kpn*I sites to generate iRFP670-PM (KRas4B tail) expression vector.

**Cell culture and transfection**. HeLa cells and HEK293T (ATCC) were maintained in Dulbecco's Modified Eagle's Medium (DMEM; PAA Laboratories GmbH) supplemented with 10% fetal bovine serum (Invitrogen) at 37 °C in a humidified 10% $CO_2$ atmosphere. Cells were transfected using either a Microporator (Neon Transfection System; Invitrogen) or Lipofectamine LTX (Invitrogen) according to the manufacturer's instructions.

**Live cell imaging**. Prepared cells were plated on 96-well polymer coverslip bottom plates (μ-Plate 96-Well ibiTreat; ibidi). R-GECO1 fluorescence imaging with blue light illumination for OptoSTIM1 activation was performed using a Nikon A1R confocal microscope (Nikon Instruments), mounted onto an inverted Eclipse Ti body (Nikon), equipped with a CFI Plan Apochromat VC objective (× 60 /1.4-numerical aperture (NA)) and digital zooming Nikon imaging software (NIS elements AR 64-bit version 3.21; Laboratory Imaging). During image processing, cells were maintained at 10% $CO_2$ and 37 °C by incubating in a Chamlide TC system (Live Cell Instruments, Inc., Korea). Immediately before imaging, the medium was replaced with OPTI-MEM (Invitrogen). Blue light photo-excitation (power density,

500 µW mm$^{-2}$) was delivered with a 488-nm laser at 5-second intervals for 1 minute, unless stated otherwise.

**Fura-2 imaging and calibration**. HeLa cells were loaded by incubating at room temperature for 30 minutes with Fura-2 AM (Invitrogen), dissolved in dimethyl sulfoxide and diluted to 2 µM in DMEM, and washed three times for 5 minutes at each step. Fura-2 imaging was performed by intermittent excitation with 340 nm and 380 nm filtered fluorescent light using a LAMBDA DG-4 lamp (Sutter Instrument Company) and a ×40//0.75 NA CFI Plan Fluor objective. The emitted light passing through a 510-nm emission filter was collected with a Nikon DS-Qi1 monochrome digital camera. Free $[Ca^{2+}]_i$ was calculated according to the formula, $[Ca^{2+}]_{free} = K_d^{Fura-2} \times (R - R_{min})/(R_{max} - R) \times F380_{max}/F380_{min}$, where $K_d^{Fura-2}$ is the dissociation constant of Fura-2 for $Ca^{2+}$, $R_{min}$ and $R_{max}$ are ratios at zero free $Ca^{2+}$ and saturating $Ca^{2+}$, respectively, and $F380_{max}$ and $F380_{min}$ are fluorescence intensities for zero free $Ca^{2+}$ and saturating free $Ca^{2+}$, respectively, at an excitation wavelength of 380 nm. According to manufacturer's instruction (Fura-2 $Ca^{2+}$ Imaging Calibration Kit (Invitrogen)), the $K_d^{Fura-2}$ in our experimental condition is set as $261 \times 10^{-9}$ M.

**InCell SMART-i assay**. HeLa cells were transfected using a Microporator (Neon Transfection System; Invitrogen) in a condition of two pulses of electric shock at 980 V for 35 milliseconds. At 12 hours post transfection, cells were treated for 2 hours with dimethyl sulfoxide (DMSO) or Rapamycin (Calbiochem), dissolved in DMSO as 2 mM stock solution and diluted to 500 nM in DMEM before use. To induce oligomerization of CRY2-fused proteins, cells were exposed to pulsatile illumination of light (5-s irradiation every 10-s, 470 nm, 100 µW mm$^{-2}$) administered with a blue LED array for 30 minutes before fixation. Cells were fixed with 4% paraformaldehyde (PFA) solution in phosphate-buffered saline (PBS) for 20 minutes and washed with PBS for three times. Cells were imaged using a confocal microscope and analyzed with NIS-element AR 64-bit version 3.21; Laboratory Imaging software provided from Nikon.

**Subjects**. All mice were handled and cared for according to the directives of the Animal Care and Use Committee of KAIST (Daejeon, Korea). All in vivo mice experiments were carried out on 8–13-week-old male C56BL/6 J mice purchased from Jackson Laboratory. Mice were housed in cages with free access to food pellets and water, and were kept on a 12-hour light–dark cycle (8 am to 8 pm) at 22 °C and 40% humidity. All behavior experiments were performed during the light phase of the light–dark cycle at the same time of day.

**Preparation of lentivirus**. The plasmid for lentiviral vectors containing *CaMKIIα* promoter was produced as previously described[6]. Plasmids for other variants were generated by cloning exchange PCR-amplified CRY2 variants (CRY2$^{E281A}$-A9, CRY2$^{D387A}$) into pLenti-*CaMKIIα*-EGFP-CRY2-STIM1 using *Age*I and *Eco*RI sites. Construct pLenti-*CaMKIIα*-EGFP was constructed by replacing ChETA-EYFP of pLenti-*CaMKIIα*-ChETA-EYFP (addgene #26967; RRID:Addgene_26967) vector to EGFP of pEGFP-C1 using *Bam*HI and *Bsr*GI sites. Plasmid containing *GfaABC1D* promoter was generated by cloning exchange PCR-amplified *GfaABC1D* promoter into pLenti-*CaMKIIα*-EGFP-CRY2-STIM1 using *Pac*I and *Bam*HI sites. The lentivirus vector was co-transfected with VSV-G and Δ8.9 required for the lentivirus production into the HEK293T cell line utilizing PEI transfection reagent. After 72 hours transfection, the supernatant was collected and centrifuged $626 \times g$ for 5 minutes and then filtered through 0.45 µM filtration unit (Millipore). For purifying lentivirus, we carried out by ultracentrifugation ($107,000 \times g$) for 2 hours at 4 °C. After ultracentrifugation, supernatant was removed and the pellet was resuspended in PBS, aliquoted and stored at −80 °C. Titration of lentivirus was measured using Lenti-X$^T$ qRT-PCR titration kit (Takara) according to the manufacturer's instructions. The viral titers were $7.88 \times 10^{11}$ and $2.23 \times 10^{12}$ genome copies ml$^{-1}$ for *CaMKIIα* promoter-bearing OptoSTIM1 and monSTIM1 viruses, respectively, and $6.81 \times 10^{11}$ and $8.42 \times 10^{11}$ genome copies ml$^{-1}$ for *GfaABC1D* promoter-bearing monSTIM1 and OptoS-TIM1(CRY2$^{D387A}$) viruses, respectively.

**Stereotaxic surgery and in vivo light-stimulation condition**. Stereotaxic viral injection was performed using 8-week-old male C57BL/6 J mice. Surgical procedures were performed under stereotaxic guidance. Before surgery, surgical tools were sterilized at 240 °C in a hot bead sterilizer. All mice, maintained at 37 °C using a temperature controller (Live Cell Instrument), were anesthetized with 0.022 ml/g Avertin and placed in a stereotaxic apparatus (Neurostar, Germany). The following coordinates (relative to bregma) were used for optical stimulation: somatosensory cortex (S1): 1.0 mm anteroposterior (AP), 2.2 mm mediolateral (ML), and −1.2 to −0.7 mm dorsoventral (DV); ACC: 1.0 mm AP, ±0.3 mm ML, and −1.0 mm DV; dorsal hippocampus (HPC): −2.0 mm AP, 1.3 mm ML and 1.2 mm (CA1)/1.8 mm (DG) DV; and thalamus (TH): −1.0 mm AP, 1.3 mm ML, and 3.4 mm DV. Lentivirus was injected at a rate of 0.075 µl/min using a 10-µl Hamilton microsyringe (Hamilton Avertin Company, USA) and a 33-gauge injection needle (NanoFil Needle Assortment, blunt; World Precision Instruments, USA). After injection, the needle was kept in place for 10 minutes before being withdrawn in stages (0.2 mm per step, 1 s each). At 4 weeks post injection, optogenetic stimulation experiments were performed using 473-nm light, delivered using a solid-state LED excitation

system (Live Cell Instrument). The duration of light illumination for observational fear learning and immunohistochemistry experiments was 30 minutes.

**Histological processing and immunohistochemistry**. Mice were anesthetized with 0.022 ml g$^{-1}$ Avertin 1 hours after light illumination (except for experiments performed under dark conditions) and perfused transcardially, first with PBS and then with 10 ml of 4% PFA in PBS. Brains were extracted and incubated in 4% PFA at 4 °C overnight. Brains were transferred to PBS, and 60-µm coronal slices were prepared using a vibratome (Leica). For immunostaining, slices were placed in PBS containing 0.2% Triton X-100 and 5% normal goat serum for 1 hour, after which the solution was replaced with primary antibody diluted in PBS containing 0.1% Triton X-100 and 2% normal goat serum. After incubating overnight at 4 °C, slices were rinsed five times with PBS containing 0.2% Tween-20 (10 minutes each), followed by a 1-hour incubation with secondary antibody. Tissue sections were then washed in PBS containing 0.2% Tween-20 and mounted on microscope slides with VECTASHIELD antifade mounting medium containing DAPI (4′,6-diamidino-2-phenylindole) (H-1200, Vector Laboratories). Fluorescence images were captured with ×10 and ×60 objectives using a Nikon A1R confocal microscope (Nikon Instruments). All ×60 images were acquired as z-stacks by binning 3-µm depths per image plane. Antibodies used for immunohistochemistry were as follows: chicken anti-GFP primary antibody (1:2000, A10262, Thermo Fisher Scientific) and Alexa 488-conjugated anti-chicken secondary antibody (1:2000, A-11039, Thermo Fisher Scientific), to stain for EGFP-CRY2(variants)-STIM1, and rabbit polyclonal anti-c-Fos primary antibody (1:1000, ab190289, Abcam) and Alexa 594-conjugated anti-rabbit secondary antibody (1:2000, A-11037, Thermo Fisher Scientific). To analyze c-Fos$^+$ cells, we quantified more than five coronal brain sections in each mice samples were used to count the number of c-Fos$^+$ cells with containing DAPI$^+$ and surrounded EGFP signal. Statistical significance was evaluated using a Sidak's multiple comparisons test.

**Observational fear**. Observational fear conditioning tests were performed in a chamber consisting of two attached two identical chambers ($18 \times 17.5 \times 38$ cm) separated in the middle of the two chambers by a transparent plexiglas divider. The floor of the cage consisted of stainless steel 5-mm rods, 1 cm apart, similar to passive-avoidance cages (Coulbourn Instruments). The space beneath the rods allowed sounds and smells to be shared during experiments. Before commencing observational fear learning experiments, mice were handled for 10 minutes over 3 days. On test day, mice were habituated to the behavior chamber for 1 hour immediately before the test. For transcranial light stimulation, the observer mouse was illuminated with blue light (1 mW cm$^{-2}$) for 30 minutes, and then was moved into the apparatus chamber containing a different mouse (demonstrator) in the next chamber. As demonstrators, 10–12-week-old mice of the same stain (C57BL/6 J) were used. In all experiments, observer and demonstrator mice were non-siblings and were housed in separate cages. Both observer and demonstrator mice were habituated to the apparatus for 5 minutes, after which a 2-second foot shock (1 mA) was delivered every 10 s for 4 minutes only to the demonstrator mouse using a programmed animal shocker (Coulbourn Instruments). On day 2 (24 hours after training), observer mice were placed in the same chamber used for observational fear learning to access contextual memory test, and were observed for 4 minutes. Fear responses were video-recorded and quantified. Data were not obtained in a blinded manner, but two other blinded investigators verified the reliability of the results. Statistical significance was evaluated using a Tukey's test.

**Pavlovian fear conditioning**. Fear conditioning experiment was performed same as what we described in previous study[6]. In brief, mice were habituated in homecage with covering LED illuminating cage lid and delivered blue light for 30 minutes prior to conditioning. Afterwards, mice were moved into conditioning chamber (Context A) consisted with shock floor (Coulbourn Instruments) to deliver electric foot shock at the bottom of chamber. Mice were habituated for 2 minutes and then gave three paired tone (30 s, 60 dB, 3 kHz)-shock (2 s, 0.7 mA) for every 2 minutes intervals. After last conditioning section, mice were left in conditioned chamber for another 90 s before moving in the homecage. The chamber was wiped sequentially with 70% ethanol and distilled water inter-training period. On day 2, mice were re-exposed into same conditioning chamber without tone presentation and recorded for 5 minutes to access the percentage of freezing behavior toward context. On day 3, mice were tested in different context which composed cylindrical shape chamber with white floor and scented with 40% diluted mouthwash (Context B). During 6 minutes recording, tone stimulation only presented at 3 minutes later after exposing the context B. Freezing behavior of mice was recorded and quantified with FreezeFrame software (Actimetrics). Two other blinded investigators verified the reliability of the results. Statistical significance was evaluated using a Tukey's test.

**Open-field test**. All mice were habituated to the testing room for 30 minutes immediately before the testing session. Mice were illuminated with blue light (473 nm) at a power density of 1 mW cm$^{-2}$ for 1 hour to activate OptoSTIM1 transcranially and sequentially transferred into the open-field box for 30 minutes test session. The total distance traveled, velocity and time spend in center were quantifying using an automated IR detection system (Optimouse)[43]. Statistical significance was evaluated using a Tukey's multiple comparisons test.

**Quantification and statistical analysis**. Images were taken and analyzed using NIS-element AR 64-bit version 3.21; Laboratory Imaging software provided from Nikon. Time measurement tool was used to quantify change of R-GECO1 intensity. Initial intensity of R-GECO1 was set as 100 and 500 (A.U.) what we previously used. The critical density of light required for activation of optoSTIM1 variants was determined based on greater than a 2-fold change in relative R-GECO1 fluorescence intensity. The maximum standard deviation of R-GECO1 intensity in cells expressing optoSTIM1 variants within 1 minute of imaging without light stimulation was 0.187973. Thus, taking a twofold change as the limit of activation, which corresponds to the limit of detection in previous studies[44], together with a basal R-GECO1 fluorescence intensity greater than 5σ, would give $< 10^{-6}$ probability of a type I error in determining activation. Measuring $T_{1/2}$ values has been previously described[30]. Statistical significance was assessed by a two-tailed Student's *t* test.

**Reporting summary**. Further information on research design is available in the Nature Research Reporting Summary linked to this article.

## Data availability

The data that support the findings of this study are available within the paper and its Supplementary Information files. Extra data are available from the corresponding author upon reasonable request.

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

## Acknowledgements

We thank all members of the laboratory for helpful discussion and comments. This work was supported by a grant from the Institute for Basic Science (no. IBS-R001-D1), KAIST Institute for the BioCentury, and KBRI basic research program through Korea Brain Research Institute funded by the Ministry of Science & ICT (19-BR-03-02), Republic of Korea.

## Author contributions

W.D.H., H.-S.S., S.L., T.K., and S.K. conceived the idea and directed the work; S.K., T.K., J.-H.C., N.K., H.P., Sehoon.K., H.M.K., S.L., H.-S.S., and W.D.H. designed experiments; S.K., T.K., N.K., and J.L. performed experiments; and S.K., T.K., S.L., N.K., J.-H.C., Sehoon.K., and W.D.H. wrote the manuscript.

## Competing interests

South Korean patent no. 10-2018-0139283 has been awarded to Institute for Basic Science (to S.K., T.K., S.L., and W.D.H., being the inventors) for the monSTIM1 technology described in this paper. The technology has been sold to Hulux and W.D.H. is a shareholder. All other authors declare no competing interests.
