## [Peer Review File · Nature Communications]

Reviewers' comments:

Reviewer #1 (Remarks to the Author):

NCOMMS-19-11686 by Kim et al. describes an improved version of OptoSTIM1, designated monSTIM1, which was developed using protein engineering of the CRY2 domain. To engineer monSTIM1, the authors designed mutants of OptoSTIM1 with mutations in a surface loop that is likely involved in self-interaction (oligomerization). This effort led to the discovery of the E281A mutant with lower basal Ca²⁺ concentration and other favourable properties. This mutation was combined with other previously reported improved CRY2 systems (CRY2olig and CRY2clust) to arrive at the monSTIM1 with the lower basal Ca²⁺ and improved photosensitivity.

The authors thoroughly characterized the performance of monSTIM1 in cultured cells, and clearly demonstrate the improvements relative to CRY2 and the other variants explored in this work. Using a custom built illumination cage, the authors demonstrate the utility of monSTIM1 in live and freely moving mice (that is, no surgically implanted fibre optics for illumination).

Overall, this is an interesting body of work that describes an improved optogenetic tool that is likely to see widespread use in biological studies of Ca²⁺ signalling. This work should be suitable for publication in Nature Communications if the following issues can be addressed.

Major comments:

1. The LED illumination chamber delivers light of 1 mW cm⁻², and the lowest light density demonstrated to activate monSTIM1 is 1 μ W mm⁻² (=0.1 mWcm⁻²). Accordingly, if my understanding is correct, the lowest level of light used to activate monSTIM1, is just 10% of the light being delivered in the light-illuminating cage. How far is this light penetrating through the hair/skin/skull? I expect that less than 10% would make it through the hair/skin/skull, and then this would continue to drop off rapidly with increasing depth into the tissue. The authors will need to discuss this critical point and include data regarding the depth at which the cells are being stimulated in the in vivo experiments. For example, for Figure 2c, does the number of c-Fos positive cells change as a function of depth into the brain? I am skeptical that sufficient light is being delivered to the hippocampus, to enable monSTIM1 activation in the fear conditioning experiment. The authors will need to provide a secondary line of evidence to convince the reader that sufficient light is being delivered.

2. Since STIM1 is itself dimeric, the rationale for monomerizing CRY2 needs to be better explained. Wouldn't the dimerization of STIM1 result in a dimeric monSTIM1, regardless of whether CRY2 is dimeric or not? One experiment that would strengthen this manuscript would be an in vitro characterization of the oligomerization of CRY2 vs CRY2 E281A. Demonstrating a change oligomeric state in the dark (and light dependent oligomerization) would help to provide a mechanistic rationale for the observed improvements.

3. Although E281A-A9 mutant has faster activation kinetics and higher activation potency, it is not apparent from Supplementary Figure 5 if this variant underwent membrane translocation as was observed in wild type CRY2. The authors will need to clarify and discuss this point, and should provide data to show that this variant does indeed undergo the expected membrane translocation.

4. The behavioural experiments with expression in the CA2 hippocampus needs to be more clearly explained. In particular, the main text and needs to be modified to be consistent with the overview schematic and the 'Context A', 'Context B', 'Cued Testing' (= tone testing?) chart labels used in the figure. I was not able to understand this section due to the inconsistencies in labeling between the figure, legend, and main text.

Minor comments:

5. The exact mutations of the CRY2clust (= A9) variant need to be defined. In addition, to avoid confusion, only one of these two names should be used consistently. For example, in the "Plasmid construction" section, both names are used individually at different places, which is confusing for

the reader. Similarly, throughout the manuscript the name monSTIM1 should consistently be used for OptoSTIM1(E281A/A9) and OptoSTIM1(CRY2E281A-A9) (perhaps written consistently as OptoSTIM1(E281A/A9) = monSTIM1).

6. It is not clear if figure 1e contains data for R-GECO1 only. This is important due to well known photoactivation of R-GECO1 when illuminated with blue light. Are the points (should be shown with X according to the legend) just obscured by other points? The authors will need to include such data and briefly discuss this concern, and to what degree it affected their results.

7. The wavelength of the blue light LED should be provided in the main text.

Grammar and presentation:

8. There are a number of strange or incorrect word choices. Some examples: "Brain utilizes" vs "The brain utilizes a"; "addresses biocompatibility" vs "introduces biocompatibility"; "alteration" vs "alternation" of basal; "distinct" vs "distinguished" behaviours. Also, the word 'proved' is a very strong term to be using for highly empirical biological work such as this, and is generally not appropriate. "Demonstrated" is a more appropriate term.

9. On page 15 under "Fura-2 imaging and calibration", it seems very odd to define "KdEGTA" as the Kd for Fura-2.

Reviewer #2 (Remarks to the Author):

Kim et al introduce an updated optogenetic tool for modulating calcium levels in neurons and glia. An upgrade from previous versions, the engineered monSTIM1 calcium channel shows internal cellular calcium levels that are on par with non-infected cells in dark (unactivated) conditions, and significantly increased internal calcium levels vs the previous version in the activated condition. This upgraded channel also shows a substantial increase in light sensitivity. Remarkably, the authors provide evidence that blue light delivered non-invasively can modulate monSTIM1-expressing cells in deep tissue (thalamus and dorsal hippocampus), and in turn can influence learning. monSTIM1 is a useful tool with several applications, from tracking molecular cascades to altering behavior. The ability to activate the channel with astonishingly small light penetration (ie, LED array on the cage lid in an intact mouse) is truly impressive. I have these comments for the authors to consider:

1) The main limitation of the study is the black box of what effects the tool is actually having on a cellular level, aside from the obvious influx of calcium. Does channel activation cause spiking of neurons (and what does this spiking look like)? Prolonged depolarization? What are the after effects, physiologically? Having a bit more clarity on these issues would allow other investigators to better interpret the usefulness/applicability of this tool. These questions would be seemingly easily addressed through intracellular or patch-clamp recordings, or perhaps with in vivo electrophysiology (even better).

2) Was c-fos only assessed in deep tissue infected with GfaABC1D (ie, glial cells)? As the behavioral results utilized the CaMKII promoter, the authors should replicate the c-fos experiment in deep tissue using this promoter, to indeed show that deep tissue pyramidal cells are affected by non-invasive blue light stimulation.

3) The authors show the construct is extremely sensitive to blue light, but how selective is its response? What is the response spectrum of monSTIM1 across different wavelengths? This will be very useful information for investigators who wish to combine this tool with other optogenetic tools.

4) Do animals need to remain in the dark to ensure no channel activation? The effects of ambient room light exposure should be examined. For example, the authors could express monSTIM1 in cortex in one experiment, and deep tissue in another experiment, and compare c-fos expression in infected vs. uninfected cells following exposure to room lighting.

5) The results of the behavioral experiments are quite clean, however it would be good to include a control group consisting of animals infected with GFP control virus (eg, no OptoSTIM) to show that OptoSTIM infection does not, in and of itself, alter learning ability. Perhaps monSTIM activation is simply restoring lost function.

MINOR:

A sentence or two of the rationale for why 30 mins of non-invasive light exposure was used (as opposed to some other duration) should be included. 30 minutes seems like a long time for Ca effects that one presumes take place on the order of single minutes. Again, having some idea of what the physiological effects of channel activation are would help with this interpretation.

The authors state, "We confirmed that light per se caused no abnormal behavior or locomotion of mice in their cage." Was this quantified in any way, or just a qualitative observation?

Point-by-point responses to the comments from Reviewers

We thank all the reviewers for their evaluation and constructive comments which profoundly enhanced our manuscript. We have addressed all the reviewer's points and carried out the requested experiments. Newly added figures are summarized as a table below. In addition, revised parts in the manuscript are designated in yellow.

Newly added figures	Description
Figure 1e	Changes in maximal R-GECO1 fluorescence intensity upon various light densities (1–1000 $\mu\text{W mm}^{-2}$).
Figure 2e,f	c-Fos stained excitatory neurons with or without monSTIM1 activation at CA1 hippocampus.
Figure 3c,d	Monitoring average freezing time during observational fear learning of mice with right ACC expressing EGFP.
Supplementary Fig. 9	Negligible effect of photoactivation of R-GECO1 in assessing Ca^{2+} influx level induced by activated monSTIM1.
Supplementary Fig. 10	Fundamental properties of monSTIM1
Supplementary Fig. 11	Analysis of oligomeric properties of STIM1-fused proteins using InCell SMART-i.
Supplementary Fig. 13e	Images showing amount of light detected in experimental conditions of either dark (ambient room light) or LED light (473 nm).

Reviewer #1 (Remarks to the Author):

We thank reviewer #1 for positive assessment of our work and constructive comments. We have made following changes on our manuscript according to reviewer #1's comments. Revised parts in the manuscript are designated in yellow.

NCOMMS-19-11686 by Kim et al. describes an improved version of OptoSTIM1, designated monSTIM1, which was developed using protein engineering of the CRY2 domain. To engineer monSTIM1, the authors designed mutants of OptoSTIM1 with mutations in a surface loop that is likely involved in self-interaction (oligomerization). This effort led to the discovery of the E281A mutant with lower basal Ca²⁺ concentration and other favourable properties. This mutation was combined with other previously reported improved CRY2 systems (CRY2olig and CRY2clust) to arrive at the monSTIM1 with the lower basal Ca²⁺ and improved photosensitivity.

The authors thoroughly characterized the performance of monSTIM1 in cultured cells, and clearly demonstrate the improvements relative to CRY2 and the other variants explored in this work. Using a custom built illumination cage, the authors demonstrate the utility of monSTIM1 in live and freely moving mice (that is, no surgically implanted fibre optics for illumination).

Overall, this is an interesting body of work that describes an improved optogenetic tool that is likely to see widespread use in biological studies of Ca²⁺ signalling. This work should be suitable for publication in Nature Communications if the following issues can be addressed.

Major comments:

1. The LED illumination chamber delivers light of 1 mW cm⁻², and the lowest light density demonstrated to activate monSTIM1 is 1 uW mm⁻² (=0.1 mWcm⁻²). Accordingly, if my understanding is correct, the lowest level of light used to activate monSTIM1, is just 10% of the light being delivered in the light-illuminating cage. How far is this light penetrating through the hair/skin/skull? I expect that less than 10% would make it through the hair/skin/skull, and then this would continue to drop off rapidly with increasing depth into the tissue. The authors will need to discuss this critical point and include data regarding the depth at which the cells are being stimulated in the in vivo experiments. For example, for Figure 2c, does the number of c-Fos positive cells change as a function of depth into the brain? I am skeptical that sufficient light is being delivered to the hippocampus, to enable monSTIM1 activation in the fear conditioning experiment. The authors will need to provide a secondary line of evidence to convince the reader that sufficient light is being delivered.

Recently, our group reported that blue light emitted from an LED (an illumination device similar to the one used in the current study) can non-invasively penetrate through the mouse hair/skin/skull and reach deep brain regions, such as hippocampus and medial septum (MS: AP 0.86 mm/ML 0.0 mm/DV 3.7 mm) (**Point-by-point Figure. 1**). We believe that monSTIM1 in deep brain regions would be also activated in a similar manner to the described tool. We included this piece of evidence in the manuscript (page 7) and referenced the paper (ref. 31).

Next, as the reviewer mentioned, we found that very low light density ($1 \mu\text{W mm}^{-2}$), which was the minimal level for stimulating the sample with our imaging device, could still activate monSTIM1, thereby inducing Ca^{2+} influx which was assessed by significant change of R-GECO1 fluorescence (**Fig. 1e**). Importantly, considering 3-fold increase of R-GECO1 intensity at light density of $1 \mu\text{W mm}^{-2}$, it is reasonable to assume that the lowest level to activate monSTIM1 could be less than $1 \mu\text{W mm}^{-2}$.

Regarding the c-Fos experiment, we provided evidence that the percentage of c-Fos positive cells decreased as a function of depth in the brain (**Fig. 2d,f**). For example, when monSTIM1 was expressed in the motor cortex, 72% of monSTIM1 positive cells were shown to be c-Fos positive. However, the percentage gradually decreased to 57% and 44% when the target sites were changed to deeper regions of dentate gyrus and thalamus, respectively. Additionally, we have newly included data in **Fig. 2e,f** where we show that monSTIM1-activated excitatory neurons in the hippocampus CA1 region elicited higher level of c-Fos expression compared to non-activated cells. Therefore, these results support our conclusion that monSTIM1 expressed in the deep brain regions is still able to respond to non-invasively-delivered blue light, owing to its superior light sensitivity.

Point-by-point Figure 1. Supplementary Figure 8 from Jung *et al.* Noninvasive optical activation of Flp recombinase for genetic manipulation in deep mouse brain regions. Nature Communications, 10 (2019) (<http://creativecommons.org/licenses/by/4.0/>). Top-down, noninvasive LED penetration and scattering patterns in the mouse brain. (a) Schematic depicting cutting of the mouse brain along a coronal plane (8-wk-old mice). (b) Side view of top-down, fiber-type blue LED (Ø 6 mm) illumination of the mouse brain. The light path was restricted from passing outside of the brain tissue by placing the fiber-type LED core at a distance of 2.5 mm from the dissected plane of the brain tissue. In addition, the skin and skull

extended longer along the sagittal axis (~1.3 mm) than the dissected plane of brain tissue, thereby thoroughly excluding scattered light from occurring at the interface between the LED fiber core and skin. (c, d) Photos of a dissected mouse brain in relation to LED position during top-down illumination in a light (c) or dark (d) environment. In d, differences in blue light distribution in mouse brain tissue between white and gray matter produced by top-down, noninvasive LED illumination are shown. Note that even gray matter (green and red arrowhead) at a depth below white matter (white arrow) shows bright penetrating light owing to the higher refractive index of white matter compared with that of gray matter. White arrow, deep cerebral white matter; green arrowhead, hypothalamus; red arrowhead, amygdala; HPC: hippocampus. Photos were taken under the same imaging conditions (focal length and exposure time) at the indicated LED intensities. Scale bar: 2mm.

2. Since STIM1 is itself dimeric, the rationale for monomerizing CRY2 needs to be better explained. Wouldn't the dimerization of STIM1 result in a dimeric monSTIM1, regardless of whether CRY2 is dimeric or not? One experiment that would strengthen this manuscript would be an in vitro characterization of the oligomerization of CRY2 vs CRY2 E281A. Demonstrating a change oligomeric state in the dark (and light dependent oligomerization) would help to provide a mechanistic rationale for the observed improvements.

As reviewer #1 mentioned, C-terminus fragment of STIM1 protein, especially SOAR (a.a 336-485) domain was known to form a dimer in the resting state (Yuan *et al.*, 2009, Yang *et al.*, 2012). Since OptoSTIM1 encompasses C-terminus of STIM1 fragment (a.a 238 – 685) along with potentially dimeric CRY2, we speculated that higher oligomeric state of OptoSTIM1 in the dark compared to that of monSTIM1 would induce elevated basal $[Ca^{2+}]_i$. To address whether CRY2^{E281A} mutation indeed results in the change of oligomeric state of monSTIM1 in the dark, we compared basal oligomeric states of OptoSTIM1 and monSTIM1 by utilizing InCell SMART-i (Intracellular supramolecular assembly readout trap for interactions, Lee *et al.*, 2011) which readily assesses protein interactions using a readout of cluster formation. We monitored efficiency of protein clustering upon rapamycin treatment in HeLa cells co-expressing FKBP-V_HH(GFP), FRB-mScarlet-FT (Ferritin) with EGFP-STIM1ct (a.a 238 – 685), OptoSTIM1 (EGFP-CRY2-STIM1ct), or monSTIM1 (EGFP-CRY2^{E281A}-A9-STIM1ct) both in the dark and light. Relatively smaller cell population with cluster formation was visualized with cells expressing either EGFP-STIM1ct (14.2%) or monSTIM1 (11%), whereas 39.2% of OptoSTIM1-expressing cell population showed cluster formation (**Supplementary Fig. 11**), suggesting that OptoSTIM1 has a higher propensity of oligomerization in the basal state than that of monSTIM1 and EGFP-STIM1ct. Notably, light illumination on both groups of cells expressing OptoSTIM1 and monSTIM1 resulted in robust and comparable level of cluster formation in the presence of rapamycin, owing to light-mediated oligomerization of CRY2. Based on this result, we suggest that CRY2 mutant (CRY2^{E281A}) would have less oligomeric property than CRY2 in the absence of blue light, thereby significantly attenuating elevated basal $[Ca^{2+}]_i$ under the overexpression condition of our optogenetic Ca^{2+} modulator.

3. Although E281A-A9 mutant has faster activation kinetics and higher activation potency, it is not apparent from Supplementary Figure 5 if this variant underwent membrane translocation as was observed in wild type CRY2. The authors will need to clarify and discuss this point, and

should provide data to show that this variant does indeed undergo the expected membrane translocation.

As reviewer #1 pointed out, we closely examined membrane translocation of monSTIM1 upon light illumination. By co-expressing plasma membrane (PM) marker, iRFP670-PM (KRas4B tail), and monSTIM1 in HeLa cells, we clearly show PM translocation of monSTIM1 upon light illumination, consistent with our previous observation with OptoSTIM1 (Kyung *et al.*, 2015). We have added the description and the data in the manuscript (page 5–6) and **Supplementary Fig. 10**, respectively.

4. The behavioural experiments with expression in the CA2 hippocampus needs to be more clearly explained. In particular, the main text and needs to be modified to be consistent with the overview schematic and the 'Context A', 'Context B', 'Cued Testing' (= tone testing?) chart labels used in the figure. I was not able to understand this section due to the inconsistencies in labeling between the figure, legend, and main text.

As the reviewer suggested, we consistently use the term 'tone' instead of cued or sound in the manuscript (page 9, 13). Thank you for the comment.

Minor comments:

5. The exact mutations of the CRY2clust (= A9) variant need to be defined. In addition, to avoid confusion, only one of these two names should be used consistently. For example, in the "Plasmid construction" section, both names are used individually at different places, which is confusing for the reader. Similarly, throughout the manuscript the name monSTIM1 should consistently be used for OptoSTIM1(E281A/A9) and OptoSTIM1(CRY2E281A-A9) (perhaps written consistently as OptoSTIM1(E281A/A9) = monSTIM1).

As the reviewer suggested, the mutation of CRY2clust (A9) is described in a more detail in the manuscript (page 4). In addition, the name of each mutant is now consistently written to avoid any confusion by readers.

6. It is not clear if figure 1e contains data for R-GECO1 only. This is important due to well known photoactivation of R-GECO1 when illuminated with blue light. Are the points (should be shown with X according to the legend) just obscured by other points? The authors will need to include such data and briefly discuss this concern, and to what it degree it affected their results.

As reviewer #1 pointed out, R-GECO1 was previously shown to possess photoactivatable property upon light illumination (Wu *et al.*, 2013, Akerboom *et al.*, 2013). In Figure 1e, we did not include the data points for cells expressing R-GECO1 without optogenetic module (Non-transfected cells were only included in Figure 1c in the initial version of manuscript), but now we have newly added data for cells expressing R-GECO1 only (Figure 1e). Thus, to examine if there was any unintended additive due to fluorescence increment of R-GECO1 regardless of monSTIM1 activity to our results, now we have included data with R-GECO1 only (**Supplementary Fig. 9**). We observed subtle increase (~1%) of R-GECO1 intensity under

repetitive illumination of blue light at 5-sec intervals for 1 minute (a typical condition for stimulating monSTIM1 in cultured cells) whereas activated monSTIM1 induced incomparably greater change (>600%) of R-GECO1 fluorescence. We also found that when pulses of blue light were transiently delivered, the increment level is shown to be only ~3%. Therefore, this result shows that the level of intensity change of R-GECO1 by blue light stimulation is negligible compared to the dynamic range of R-GECO1 by monSTIM1 activation that we are mostly looking at

7. The wavelength of the blue light LED should be provided in the main text.

We wrote the wavelength of blue light illuminated by LED as 473 nm on the revised manuscript (Page 7).

Grammar and presentation:

8. There are a number of strange or incorrect word choices. Some examples: “Brain utilizes” vs ‘The brain utilizes a’; “addresses biocompatibility” vs “introduces biocompatibility”; “alteration” vs “alternation” of basal; “distinct” vs “distinguished” behaviours. Also, the word ‘proved’ is a very strong term to be using for highly empirical biological work such as this, and is generally not appropriate. “Demonstrated” is a more appropriate term.

We thank reviewer for the consideration and have corrected grammars presented at main texts.

9. On page 15 under “Fura-2 imaging and calibration”, it seems very odd to define “KdEGTA” as the Kd for Fura-2.

We thank reviewer #1 for pointing out our mistake. The definition of K_d^{EGTA} is corrected as the dissociation constant of EGTA for Ca^{2+} . As measurement (Quantification) of absolute intracellular Ca^{2+} concentration using Fura-2 AM dye was achieved according to manufacturer’s guidance, this mistake does not change any of our results.

Reviewer #2 (Remarks to the Author):

We thank reviewer #2 for the constructive remarks. According to those comments, we have made following changes and improved the manuscript. Revised parts in the manuscript are designated in yellow.

Kim et al introduce an updated optogenetic tool for modulating calcium levels in neurons and glia. An upgrade from previous versions, the engineered monSTIM1 calcium channel shows internal cellular calcium levels that are on par with non-infected cells in dark (unactivated) conditions, and significantly increased internal calcium levels vs the previous version in the activated condition. This upgraded channel also shows a substantial increase in light sensitivity. Remarkably, the authors provide evidence that blue light delivered non-invasively can modulate monSTIM1-expressing cells in deep tissue (thalamus and dorsal hippocampus), and in turn can influence learning. monSTIM1 is a useful tool with several applications, from tracking molecular cascades to altering behavior. The ability to activate the channel with astonishingly small light penetration (ie, LED array on the cage lid in an intact mouse) is truly impressive. I have these comments for the authors to consider:

1) The main limitation of the study is the black box of what effects the tool is actually having on a cellular level, aside from the obvious influx of calcium. Does channel activation cause spiking of neurons (and what does this spiking look like)? Prolonged depolarization? What are the after effects, physiologically? Having a bit more clarity on these issues would allow other investigators to better interpret the usefulness/applicability of this tool. These questions would be seemingly easily addressed through intracellular or patch-clamp recordings, or perhaps with in vivo electrophysiology (even better).

In the previous study (Kyung *et al.*, 2015), we observed that CRAC channel activated by OptoSTIM1 selectively allows Ca²⁺ influx but not Na⁺ current (**Point-by-point Figure 2a**). In addition, we characterized electrophysiological properties of CA1 hippocampal excitatory neurons expressing OptoSTIM1. Unlike Channelrhodopsin (ChR2), OptoSTIM1 did not evoke action potential during 25 minutes of whole cell patch-clamp recording (**Point-by-point Figure 2b,c**). Interestingly, even though the changed value was not significant and sample size was small ($n = 3$ neurons from 2 mice), we observed a tendency of slight increase of resting membrane potential upon OptoSTIM1 activation.

In order to further examine if there would be any potential influence from our optogenetic calcium modulator on electrical properties of neurons, we have tried to perform whole cell patch-clamp recording with brain slices where excitatory neurons in the ACC expressed monSTIM1. Similar to what we observed previously, monSTIM1-positive pyramidal neurons did not evoke action potential, but two cells among six showed noticeable increase of resting membrane potential after 5 minutes of continuous exposure to blue light. Yet, in the quantified result, we could not see significant difference of this change with the baseline. Unfortunately, during the experiment, we realized that it was hard to maintain whole cell configuration for longer than 5 minutes, so it is unclear whether the changed membrane potential is solely attributed by monSTIM1 activation. Nevertheless, we fully agree with the reviewer's comment that it is surely important to deeply understand effects of monSTIM1 on the cell physiology in

many aspects. But at this moment, since our data is too preliminary to be presented in this work, we think that we need to further optimize the experimental condition to characterize physiological facets of monSTIM1 in a more detail in the future study.

Reprinted by permission from Springer Nature: Nature, Nature Biotechnology. Optogenetic control of endogenous Ca²⁺ channels in vivo, Kyung et al. © 2019 Springer Nature Limited (2015).

Point-by-point Figure 2. (a) Current-voltage (I-V) relationships of CRAC currents (Fig 1c from Kyung et al., 2015). HEK293 cells were undergone whole cell patch-clamp by ramping membrane potential from -100 mV to +80 mV either before (green line) or during (red and blue lines) OptoSTIM1 activation. External medium includes 10 mM Ca²⁺, and to assess contribution of Na⁺ current, Na⁺ in external medium was replaced with NMDG⁺. (b-d) Electrophysiological recording of hippocampal neurons in brain slices (Supplementary Figure. 20 from Kyung et al., 2015). (b) Representative graph showing membrane potential upon OptoSTIM1 activation for 25 min (blue bar) in a CA1 hippocampal neuron. Graph showing average membrane potential during 1 min either before (-) or after (+) the OptoSTIM1 stimulation. Grey line indicates values from each cell, and black line indicates average values. $n = 3$ neurons from 2 mice. Error bars, s.e.m. (c) Response pattern upon -100 pA and +200 pA current injection after experiment in a. (d) Representative graph showing response curve of membrane potential according to ChR2 activation (left). The graph on the right represents expanded view of burst of action potentials shown on the left graph. Blue bars indicate time points of illumination.

Point-by-point Figure 3. Recording resting membrane potential (RMP) of the cells expressing monSTIM1 at ACC **(a)** Representative graphs showing resting membrane potential of cells before (upper) and photoactivation for 5 minutes (below) in ACC. **(b)** Graph indicating average RMP (mV) either before or after blue light irradiation. Individual value of cells were marked as dots. $n = 6$ neurons. NS = Not significant by Student's two-tailed t -test. Error bars, s.e.m.

2) Was c-fos only assessed in deep tissue infected with GfaABC1D (ie, glial cells)? As the behavioral results utilized the CaMKII promoter, the authors should replicate the c-fos experiment in deep tissue using this promoter, to indeed show that deep tissue pyramidal cells are affected by non-invasive blue light stimulation.

We agree with the reviewer's point that we should show the increased c-Fos expression in excitatory neurons expressing monSTIM1 in deep tissue, such as hippocampus, for the consistent flow of our data. To address the issue, we injected *CaMKII α* -driven monSTIM1 at CA1 hippocampus for the c-Fos experiment. With the previously described experimental condition (30 minutes non-invasive light illumination at 1 mW cm⁻² light density), light irradiation efficiently induced c-Fos expression in 21.5% of monSTIM1 expressing CA1 hippocampal excitatory neurons, whereas only 1.1% of c-Fos-positive neuronal cell population was observed in non-stimulated mice, reflecting that non-invasive illumination of light could activate monSTIM1, thereby inducing c-Fos expression. We have included these results and detailed information of statistical analyses in the result section of revised manuscript (page 8, **Fig. 2e-f**).

3) The authors show the construct is extremely sensitive to blue light, but how selective is its response? What is the response spectrum of monSTIM1 across different wavelengths? This will be very useful information for investigators who wish to combine this tool with other optogenetic tools.

As the reviewer pointed out, we have carried out experiment in which we co-expressed monSTIM1 and R-GECO1 in HeLa cells and monitored relative fluorescence intensity of R-GECO1 upon different wavelengths of light illumination (405, 457, 488, 514, 561 and 640 nm). MonSTIM1 efficiently responded to 457 or 488 nm light, but weakly or not responded to the 405 nm light and wavelength of light longer than 514 nm (page 6, **Supplementary Fig. 10d**).

4) Do animals need to remain in the dark to ensure no channel activation? The effects of ambient room light exposure should be examined. For example, the authors could express monSTIM1 in cortex in one experiment, and deep tissue in another experiment, and compare c-fos expression in infected vs. uninfected cells following exposure to room lighting.

We apologize the reviewer for confusion on use of terminology. In fact, the condition we designated as "the dark state" represents ambient room light environments. To further address the reviewer's concern, we measured the intensity of blue light (473 nm) in the LED chamber with two different conditions side-by-side (room light only and room light + LED). The light density of room light only condition was about 1.8 μ W cm⁻² whereas light density of illumination by LED exhibited 1000 μ W cm⁻² (**Supplementary Fig. 13e**). Based on the control experiments for c-Fos staining and mouse behaviors (Dark controls in **Fig. 2 and 3**), we believe the ambient room light exposure in our experimental condition would hardly affect monSTIM1

activities. To avoid any confusion, we have described the “dark” condition more precisely for mouse experiments in the revised manuscript.

5) The results of the behavioral experiments are quite clean, however it would be good to include a control group consisting of animals infected with GFP control virus (eg, no OptoSTIM) to show that OptoSTIM infection does not, in and of itself, alter learning ability. Perhaps monSTIM activation is simply restoring lost function.

To address whether viral infection itself could affect learning ability of mice, we generated *CaMKII α* promoter-driven EGFP lentivirus and expressed the gene in mice brain. Mice injected by either monSTIM1- or EGFP-expressing virus at right ACC were tested for observational fear learning as we previously described. Consistent with the previous result, we were able to observe a substantial increase of vicarious freezing response in the monSTIM1 expressing mice exposed with blue light (473 nm, 1 mW cm⁻²) for 30 min ($n = 4$). However, EGFP expressing mice ($n = 10$) showed no significant difference in terms of freezing compared to mice expressing monSTIM1 without light illumination ($n = 4$), demonstrating that infection of monSTIM1-encoding virus itself did not alter observational fear response and learning. We included additional behavioral results and detailed information of statistical analyses in the results section of revised manuscript (page 8, **Fig. 3c–d**).

MINOR:

A sentence or two of the rationale for why 30 mins of non-invasive light exposure was used (as opposed to some other duration) should be included. 30 minutes seems like a long time for Ca effects that one presumes take place on the order of single minutes. Again, having some idea of what the physiological effects of channel activation are would help with this interpretation.

We selected light illumination time as 30 minutes based on several experimental tests we have previously done (unpublished results). Representatively, significant effects on behavior of mice expressing OptoSTIM1 at CA1 hippocampus were only observed when mice were illuminated with blue light more than 25 minutes (**Point-by-point Figure 4**). In contrast, blue light stimulation for 5 minutes either 20 minutes before or right before the conditioning does not alter the freezing level compared to non-light stimulated mice. Therefore, this result implies that there might be a certain threshold of time window for monSTIM1 activation to produce significant behavioral outcome. The identity of threshold might be the number of responded cells or activation of a set of certain Ca²⁺-responsive molecular machineries that require persistent increase of Ca²⁺ for at least 25 minutes rather than timescale of seconds or a few minutes. Along with our future plan to characterize electrophysiological property induced by monSTIM1 activity, we are currently studying this issue in a more detail that will further expand the utility of monSTIM1 and can be developed into a new perspective on Ca²⁺ mediated physiological effects.

Point-by-point Figure 4. (a) Schematic figure described *Pavlovian* fear conditioning. Prior to conducting fear conditioning, mice expressing OptoSTIM1 at CA1 hippocampus were illuminated by blue light (473 nm, 7 mW, 20 Hz) through optic fiber in designated condition (0 min, 5 min Light On, 5 min Light On – 20 min Light Off, 25 min Light On). After 2 min of habituation at context A, a foot shock was applied for 2 sec at the end of 30 sec tone delivery. **(b – c)** Contextual fear memory test at context A 24 hour post training * $P < 0.05$, Student’s two-tailed t -test. Error bars, s.e.m.

The authors state, “We confirmed that light per se caused no abnormal behavior or locomotion of mice in their cage.” Was this quantified in any way, or just a qualitative observation?

We did not quantify behavioral changes of mice at their homecage before and after light illumination by customized LED system. To avoid any misinterpretation, we changed the sentence in the revised manuscript (page 7).

REVIEWERS' COMMENTS:

Reviewer #1 (Remarks to the Author):

For the most part, the authors have addressed my comments to my satisfaction. However, it seems that they may have misunderstood point #9 in my original review:

9. On page 15 under "Fura-2 imaging and calibration", it seems very odd to define "KdEGTA" as the Kd for Fura-2.

The author's now write "[Ca²⁺]_{free} = Kd EGTA × (R – R_{min})/(R_{max} – R) × F380_{max}/F380_{min}, where Kd EGTA is the dissociation constant of EGTA for Ca²⁺". This is incorrect. The free Ca²⁺ concentration must, of course, be calculated using the Kd for Fura-2, rather than the Kd for EGTA. I believe that the confusion is caused by a typo in equation 3 in the instructions included with Fura-2 Ca²⁺ Imaging Calibration Kit (Invitrogen). Equation 3 should be Kd(fura-2), not Kd(EGTA). The equations in the original manuscript (The Journal of Biological Chemistry, 1985, 260, 3440-3450) are correct.

I also noticed that the color bar for Figure 1g does not match the colors in the figure (that is, there is no blue in the color bar...)

Once these issues have been addressed, the manuscript should be suitable for publication.

Reviewer #2 (Remarks to the Author):

My concerns were satisfactorily addressed, the inclusion of new results has strengthened the paper and further support the conclusions proposed by the authors. I only have a few minor comments

The weakness of the paper remains the unknown physiological/molecular mechanisms driving the observed cellular (increased c-fos) and behavioral (fear conditioning) effects. Given the considerable increase in internal calcium concentration on such a short time scale (~8x increase in [Ca] over the course of one minute; Fig S7), it is very surprising that this does not translate to a significant change in membrane potential. Is it possible that other ions may be transported through this channel? The authors previously show that Na⁺ is not being transported, but what about other ions that may act to offset Ca influx? Can the authors provide any insight into this? Although the lower-level mechanism of action is of high importance to fully interpret the function and higher-level effects of monSTIM1 expression, I concede they are beyond the scope of the current study. However a few lines of discussion may be helpful.

Please provide the rationale in the manuscript for 30 min of non-invasive light exposure was used for experiments.

When explaining the potential reasons as to why effects were only seen following 25+ minutes of light stimulation, the authors' state "The identity of threshold might be the number of responded cells or activation of a set of certain Ca²⁺-responsive molecular machineries that require persistent increase of Ca²⁺ for at least 25 minutes rather than timescale of seconds or a few minutes." Citing references in support of either speculation would greatly strengthen this argument, and should be included in the Discussion.

The use of the term 'Dark' to refer to animals exposed to ambient light is misleading (and the clarification is only briefly referred to in the Discussion section). Please either make this distinction explicit earlier in the manuscript, or better yet, rename 'Dark' to something more accurate, such as 'Ambient.'

Point-by-point responses to the comments from Reviewers

We thank all the reviewers for their evaluation and constructive comments which profoundly enhanced our manuscript. We have addressed all the reviewer's points and described in the revised manuscript.

Reviewer #1 (Remarks to the Author):

For the most part, the authors have addressed my comments to my satisfaction. However, it seems that they may have misunderstood point #9 in my original review:

9. On page 15 under “Fura-2 imaging and calibration”, it seems very odd to define “KdEGTA” as the Kd for Fura-2.

The author’s now write “[Ca²⁺]_{free}=Kd EGTA × (R – R_{min})/(R_{max} – R) × F_{380max}/F_{380min}, where Kd EGTA is the dissociation constant of EGTA for Ca²⁺,”. This is incorrect. The free Ca²⁺ concentration must, of course, be calculated using the Kd for Fura-2, rather than the Kd for EGTA. I believe that the confusion is caused by a typo in equation 3 in the instructions included with Fura-2 Ca²⁺ Imaging Calibration Kit (Invitrogen). Equation 3 should be Kd(fura-2), not Kd(EGTA). The equations in the original manuscript (The Journal of Biological Chemistry, 1985, 260, 3440-3450) are correct.

I also noticed that the color bar for Figure 1g does not match the colors in the figure (that is, there is no blue in the color bar...)

Once these issues have been addressed, the manuscript should be suitable for publication.

We thank reviewer #1 for pointing out our misunderstanding. Now we corrected Kd^{EGTA} into Kd^{Fura-2} in the equation based on the original publication the reviewer referenced. In addition, we correctly put matching color bars in **Figure 1g**. Thanks for thorough examination on our figures.

Reviewer #2 (Remarks to the Author):

My concerns were satisfactorily addressed, the inclusion of new results has strengthened the paper and further support the conclusions proposed by the authors. I only have a few minor comments

The weakness of the paper remains the unknown physiological/molecular mechanisms driving the observed cellular (increased c-fos) and behavioral (fear conditioning) effects. Given the considerable increase in internal calcium concentration on such a short time scale (~8x increase in [Ca] over the course of one minute; Fig S7), it is very surprising that this does not translate to a significant change in membrane potential. Is it possible that other ions may be transported through this channel? The authors previously show that Na⁺ is not being transported, but what about other ions that may act to offset Ca influx? Can the authors provide any insight into this?

Although the lower-level mechanism of action is of high importance to fully interpret the function and higher-level effects of monSTIM1 expression, I concede they are beyond the scope of the current study. However a few lines of discussion may be helpful.

Please provide the rationale in the manuscript for 30 min of non-invasive light exposure was used for experiments.

When explaining the potential reasons as to why effects were only seen following 25+ minutes of light stimulation, the authors' state "The identity of threshold might be the number of responded cells or activation of a set of certain Ca²⁺-responsive molecular machineries that require persistent increase of Ca²⁺ for at least 25 minutes rather than timescale of seconds or a few minutes." Citing references in support of either speculation would greatly strengthen this argument, and should be included in the Discussion.

The use of the term 'Dark' to refer to animals exposed to ambient light is misleading (and the clarification is only briefly referred to in the Discussion section). Please either make this distinction explicit earlier in the manuscript, or better yet, rename 'Dark' to something more accurate, such as 'Ambient.'

We thank the reviewer #2 for giving us constructive comments. Regarding the ion selectivity, CRAC channels are reported to possess highly distinguished Ca²⁺ ion selectivity in numerous papers (Representatively, McNally *et al.*, 2012). Also, in terms of consequential effects on the membrane potential upon CRAC channel opening, it has been implicated in the previous literature that K⁺ ions are rapidly pumped out from CRAC channel-opened cells through Ca²⁺-activated K⁺ pumps to maintain Ca²⁺ influx, in which case the membrane potential is not affected (Cahalan *et al.*, 2007). Also, as reviewer#2 mentioned, we provided additional

rationales in the discussion section of our manuscript as to why prolonged light exposure (>25 min) was utilized in our experiments. We addressed two probable reasons for the phenomenon, one at transcriptional level and the other one at post-translational level. First, it has been implicated in previous literatures that endogenous c-fos expression is a metrics to assess neuronal activity and it reaches a maximum level between 30 to 40 minutes post Ca^{2+} signaling-triggering stimulus (Garner *et al.*, 2012). In terms of cell population, it was reported that the number of c-fos positive cells is positively correlated with the level of behavioral outputs such as freezing. Although identifying the time point where endogenous c-fos expression is saturated in neuronal cells upon monSTIM1 activation is of further research, we highly suspect that approximately 30 min of monSTIM1 stimulation is sufficient to achieve the level of neuronal activity and cell population that can robustly translate Ca^{2+} -dependent activation of molecular machineries into changes in behavioral phenotypes, enhanced social fear learning in this case. Another speculation is that prolonged Ca^{2+} influx changes post-translational context of synapses via activity-induced synaptic remodeling molecules, such as GluR and CaMKII, which are previously shown to influence behavioral effects (Kennedy, *et al.*, 2010; Pisansky *et al.*, 2017; Lisman *et al.*, 2012. Saucerman and Bers, 2008) upon their modulated activities. In addition, it has been recently shown through chemogenetic tools that AMPA and NMDA receptor activities can be gradually modulated over the course of 30 min post ligand stimulus in hippocampal excitatory neurons (Pati *et al.*, 2019), indicating that Ca^{2+} -responsive molecular machineries may take time scale of minutes to impose their influences on higher level phenotypes. We have described the points in the discussion section and cited appropriate references.

Moreover, to avoid confusion with the term ‘Dark’ *in vivo* experiment, we changed it into ‘Ambient’ and described it in our main text.